# Enhancing semantic belief function to handle decision conflicts in SoS using k-means clustering

Eman K. Elsayed[1,2,*], Ahmed Sharaf Eldin Ahmed[3,4] and Hebatullah Rashed Younes[3,*]

[1] Mathematical and Computer Science Department, Faculty of Science, Al-Azhar University (Girls Branch), Cairo, Egypt
[2] Computer Science Department, Faculty of Information Technology, Misr University for Science and Technology (MUST), Giza, Egypt
[3] Information Technology Department, Faculty of Information Technology and Computer Science, Sinai University, Arish, Egypt
[4] Information Systems Department, Faculty of Computers and Artificial Intelligence, Helwan University, Cairo, Egypt
* These authors contributed equally to this work.



Corresponding author
Hebatullah Rashed Younes,
heba.rashed@su.edu.eg

## ABSTRACT

**Background:** The endeavouring to offer complex special functions from individual systems gave rise to what is known as the System of Systems (SoS). SoS co-integrating systems together while allowing for absorbing more systems in the future. SoS as an integrated system simplifies operations, reduces costs, and ensures efficiency. However, conflict may result while co-integrating systems, violating the main benefits of SoS. This paper is concerned with enhancing the time required to detect and solve such conflicts.

**Methods:** We adopted the k-means clustering technique to enhance the detection and solving of conflict resulting while co-integrating new systems into an existing SoS. Instead of dealing with SoS as a single entity, we partition it into clusters. Each cluster contains nearby systems according to pre-specified criteria. We can consider each cluster a Sub SoS (S-SoS). By doing so, the conflict that may arise while co-integrating new systems can be detected and solved in a shorter time. We propose the Smart Semantic Belief Function Clustered System of Systems (SSBFCSoS), which is an enhancement of the Ontology Belief Function System of Systems (OBFSoS).

**Results:** The proposed method proved the ability to rapidly detect and resolve conflicts. It showed the ability to accommodate more systems as well, therefore achieving the objectives of SoS. In order to test the applicability of the SSBFCSoS and compare its performance with other approaches, two datasets were employed. They are (Glest & StarCraft Brood War). With each dataset, 15 test cases were examined. We achieved, on average, 89% in solving the conflict compared to 77% for other approaches. Moreover, it showed an acceleration of up to proportionality over previous approaches for about 16% in solving conflicts as well. Besides, it reduced the frequency of the same conflicts by approximately 23% better than the other method, not only in the same cluster but even while combining different clusters.

## INTRODUCTION

Recently, and according to systems, it has been proven that in order to create a powerful, co-integrated, and multitasking system, an individual system will not be sufficient (*Boehm & Lane, 2007*; *Robinson, Pawlowski & Volkov, 2003*; *Viana, Zisman & Bandara, 2017*). Therefore, to achieve such a system, some systems must be co-integrated together in a System of Systems (SoS) (*Spanoudakis & Zisman, 2001*). SoS is a set of dedicated systems that have fetched their abilities to create a new complicated system, providing more performance and services than individual systems (*Popper et al., 2004*). Organizations are continually facing challenges to co-integrate new Component Systems (CS) and update existing systems while under threats, restricted budget, and uncertainty (*Agarwal, Dagli & Pape, 2016*). Both complexity and uncertainty are inherent features of infrastructure SoS, which can drive the operation of such systems away from their intended purposes (*Peculis & Shirvani, 2017*).

It is necessary for a successful system to accurately define the interests, objectives, and requirements of its components, therefore, there are some limitations. SoS has been facing different challenges due to the configuration and functions of the systems that make SoS up. One of such challenges is managing goals to make an appropriate decision on SoS.

CS that participates in SoS arrangements might cause conflicting individual objectives among themselves. As well as, the emergence of conflicting objectives between the entire SoS, it is CS (*Sage, 2003*).

Our objective is to have more CS within a single SoS; however, such systems must be harmonious. A conflict is an issue that occurs between two or more systems in SoS, and it represents a fundamental issue that has been extensively discussed for SoS. Such an issue might be security, pattern, classifications, or decision. Resolving the conflict does not occur through one stage but more. First, identify and detect the conflict, then diagnose to fix it appropriately (*Ramsbotham, Miall & Woodhouse, 2011*).

Multi-systems conflicts are a generalisation of a single systems conflict problem for more than one system, hence, in SoS, the more CS, the more conflicts. Several algorithms have been proposed to deal with conflicts among CS in SoS, of which most deal with any CS in SoS as a single system. In such cases, the aim is finding a conflict resolution for different systems, guaranteeing that the given solution is optimal, and managing to handle a specified number of CS.

This paper aims to find a better solution to the conflict problem. Better solutions are usually applied when SoS CS are individuals. The mission is to attract and contain more SoS and get better conflict resolution out of treating SoS CS as a joint system in clusters, and in turn, the concept of clustering, and time reduction is important from a semantic point of view.

Clustering is a learning method that organises objects that are with similarities in one or more features into clusters (*Mokhtarpour & Stracener, 2014*), aiming to isolate objects with similar traits and assign them into clusters. Consequently, eventually, each cluster should include identical items which are different from those of other clusters.

Clustering algorithms are unsupervised pattern-learning algorithms (*Wu et al., 2020*) without prior information, through creating smaller clusters with high intergroup dissimilarities and intragroup similarities. A large number of these algorithms found. Hierarchical, Partitional, and Bayesian are popular ones. This kind was purposefully chosen since SoS is a learned-system, and unsupervised learning as a learning technique has permitted the model to work with its discovery information; mainly deals with unclassified data.

The proposed approach aimed to adopt as many CS as possible into clusters, and to, get such clusters to form a significant SoS. Successful harmonisation of CS together in an SoS requires quick and appropriate handling of conflicts.

There are some contributions to this paper as follows:

- Introduction of a new method to add more systems to SoS.
- Improving the handling techniques of the issues especially conflicts.
- Illustration of the potential of the proposed method to group more systems into clusters in what resembles Sub SoSs (S-SoSs) within a large main SoS named General SoS (G-SoS).
- Demonstration of how the clustering method was more appropriate to cover objectives and influence results, that is not in attracting more systems, but rather in reducing the number of conflicts, which has, in turn, led to more time reduction enhanced the overall results.

This paper is organised as follows;

Part 1 covers a brief review of the conflicts in SoS, categorises all existing work into some main categories (Decision, design, security and code), orders the different approaches for solving this problem, then, in turn, displays some clustering techniques with related works. The main focus was on the method, which has been developed for use.

Part 2 presents a way to attract more system and optimise conflict resolution in SoS, explains the new algorithm (SSBFCSoS) that divided into two parts (levels). The single CS has initialised with their default (working) task, which may contain conflicts, then use k-means clustering to make them S-SoSs for the G-SoS.

## BACKGROUND AND RELATED WORKS

The System of Systems (SoS) includes complex components and complex calculations (*Gorod et al., 2014*), with challenges of which the most significant are design, development, and decision. The detailed intricate design of SoS requires managing its CS to take advantage of it, which in turn necessitates CS interoperability rather than compatibility with diverse assets (*Luzeaux, 2014*; *Lane & Boehm, 2019*), which does not usually fit the operational needs of the large SoS. CS of SoS must be able to exchange data or information among themselves, either for its purposes or for integration and fusion among them, and eventually the exchange process between the architecture of different systems, that may cause design conflicts.

A survey in 2010 (*Engineering, 2010*) explained the information and construction of SoS from an engineering standpoint, combining various designs of CS that often cause conflicts. The methods used often create conflicts as well, and clarify how data managed between CS and how to face conflicts, through discovering another source of data in the event of a conflict. Such process presented a new conflict if no other source was available, rather than a solution to the conflict, contrary to what our research paper has provided, as in using BF to review source reliability, depending upon the trusted source system.

Various security priorities of multiple CS in SoS can cause conflicts (*Bodeau, 1994*), therefore, the available security mechanisms may not meet specific SoS security requirements, leading to a more complex task of integrating and architecture SoS (*Maier, 1998*). If a particular system has to interact securely with another CS system, its security architecture may have to deal with different authentication mechanisms, presenting a security challenge in the SoS.

Some security issues of CS that make-up SoS have been discussed according to the characteristics of SoS (*Madan, 2015*), among which is integrating the different security structures of the CS (*Chiprianov et al., 2014*), which requires high characteristics of the SoS to maintain the system, and guaranteeing the services despite some security problems, in terms of networks and others.

The research (*Ki-Aries et al., 2018*) discussed an approach that coordinates the concepts of SoS for analysing security conflicts, depending on the independent collaboration of CS in SoS. The research has explained that unknown information elements have frequently impeded the risk assessment and decision-making regarding SoS. Contrary to our research, where the BF equation has been used for this, to verify that—assumed unknown—information, which has enabled us to make an appropriate decision for SoS, regardless of security conflicts either in this or other research.

The conflict-resolution algorithm has been introduced for multi-agent systems related to relationships of different types and values (*Garanina, Sidorova & Anokhin, 2015*) to obtain conflict-free factors and solve ontology-based ambiguity in analysing the natural language text. The algorithm has relied on an agent weighting conflict resolution. Contrary to our research, which has relied on the decision weight and both its accuracy and reliability, leading to more comfortable and faster processing of complex systems since it depends on addressing the conflicting data rather than the system or agent.

For systems containing databases, the time factor is essential, therefore as per the research (*Ahmed, Salem & Saleh, 2015*), more time may be needed to obtain data from the site, especially from a distance. The process of clustering has helped to address this challenge. The concept was reducing a large number of databases through clustering, as proposed by the research, for both distribute data and systems, reducing the number of conflicts within SoS, and reducing the mass conflict, according to the reliability estimate. It includes performance improvement and decision-making while saving time.

Researchers (*Wu et al., 2020*) have proposed a clustering algorithm evaluation model to assess conflicting performance, to reconcile the differences in the evaluation performance of these algorithms. These conflicts have been shown during the process of

integrating the information during the decision-making process. Moreover, the proposed model can reconcile conflicting evaluation performance to reach a collective agreement in the complex decision-making environment.

Researchers have used the algorithm k-means to reduce the data dimensions (*Idrees & Gomaa, 2020*), proposing a two pillars method. The first pillar is to define the consistency of traits more precisely, as in the aspect of applying clustering techniques to remove less likely features.

Based on the kernel k-mean clustering algorithm, a proposed study (*Muflikhah et al., 2020*) has shown that the algorithm could detect disease. Two types of datasets were extracted from DNA lysis and grouped into two groups using the kernel k-mean algorithm, then the classifier was applied to each group. When datasets aggregated, each group has significant similarities in characteristics. The block dependent classification contains more support vectors than the no-group category. Block quality also affects detection performance.

# CLUSTERING AND ITS TECHNIQUES

Clustering is a significant unsupervised learning problem, which draws references from datasets consisting of input data without specific responses, and is used as a process to find meaningful structures (*Rokach & Maimon, 2005*).

There are no specific criteria for exemplary assembly, however, it depends on the appropriate standards for users to meet their needs. For instance, in our method, we have worked on finding representatives for Homogeneous Groups for better performance and adopted clustering to reduce working all systems together. Contrary to our approach, the previous techniques have detected conflicts on all CS of SoS, while we have used clustering to divide these CS into small clusters, leading to not only more systems attraction, but better performance and conflict resolution.

## Clustering methods are divided into three parts

a) **Hierarchical:** The algorithms of this technique depend on finding its clusters using pre-created ones. It can be either agglomerative or divisive (*Rokach & Maimon, 2005*). Agglomerative (bottom-up) starts with each object as a separate cluster and then merges them into larger clusters while Divisive (top-down) begins with the whole set, then divides it into smaller clusters.

b) **Partitional:** The algorithms here define all clusters then use the Divisive algorithms technique for grouping. It contains many techniques as Model-Based, Graph-Theoretic, and Spectral. Besides, Centroid techniques to which the k-means algorithm belongs. Clustering deals mostly with data segmentation within mega systems. Contrary to our research paper, which depends on dividing the systems within SoS rather than the data, the k-mean technique has been used for the clustering process. As a partitional algorithm, it starts with a fixed number of clusters. After determining the number of clusters, the class of object $n$ is determined (*Rokach & Maimon, 2005*), then assigned to the nearest cluster centre. The centres of mass $k$ are then re-estimated assuming that

the above memberships are valid. This process is repeated until $n$-objects membership does not change in the last iteration.

c) **Bayesian:** The algorithms of this technique generate posterior distribution to collect all data partitions. It has two algorithms decision Based and Nonparametric.

# SMART SEMATIC BELIEF FUNCTION CLUSTERED SYSTEM OF SYSTEMS METHOD (SSBFCSOS)

The ability of SoS to perform large tasks lies in coordinating its CS as a single system. The more the SoS can assemble more systems and do this coordination, the better. Obtaining the optimum decision from all systems while maintaining security is the goal of SoS. SoS is constructed from many CS, which conflicts among them cause SoS to be inefficient. In 2019, researchers (*Younes, Ahmed & Elsayed, 2019*) have adopted a method to get the best Decision for SoS while avoiding conflicts between CS. The algorithm used was ontology with the Belief Function equation that modified to suit systems. Learning has limited SoS capacity to obtain the best results, since during running, and when CS reaches a specified number, SoS gets more time to handle conflicts. As a learned-system, it represents a challenge, so, an SSBFCSoS algorithm that depends on the clustering idea was suggested.

## The handled clustering technique for SoS in SSBFCSoS

Clustering is the idea for SSBFCSoS to divide CS, not data into clusters. We have primarily relied on the use of the k-means method, and a combination of this method and another agglomerative method (bottom-up). According to the used k-means method; defining each cluster according to its properties requires creating clusters that share common properties. We adopted an idea that depends on the location of the CS systems, hence each cluster contains nearby systems according to pre-specified criteria.

While in the case of partitional clustering k-means, the $k$ partitions created containing $n$ objects, each partition represents a cluster created, where $k \leq n$, attempting to divide items into clusters based on some evaluation criteria. A specific number of clusters $k$ determined by a set of centroids (initially assigning a centroid to each cluster) where $C = \{C_1, \ldots, C_k\}$. The centroid calculation for each cluster according to the mean $M_k$ of all the objects $N_k$ that belong to this cluster as in Eq. (1).

$$M_k = \frac{1}{N_k} \sum_{q=1}^{N_k} X(q) \tag{1}$$

The members define each centre. Moreover, if the properties are incompatible, re-merge them again with any other appropriate cluster. Furthermore, this matter is repeated until each object belongs to a suitable cluster. Moreover, the stability of the clusters and their centres occurs without change. Each cluster is to be considered as Sub SoS (S-SoS) that contain systems as objects.

Determining the appropriate S-SoS to belong to a particular system depends on how close the traits are between systems of the same S-SoS. The primary is the close locations

between systems in each of S-SoS. The k-means algorithm measures the extent of similarity by calculating distance. Both distance and similarity are related. Distance measure determines the similarity between items and affects the shape of clusters, a smaller distance between two objects results in more similarity.

Initialising S-SoSs is achieved by picking one point per the first S-SoS randomly as an initial centroid point, then $k - 1$ for the other S-SoSs; each centroid point is as far away as possible from the previous centroid ones.

Depending on the k-means method (*Solanki & Pittalia, 2016*), the distance is calculated between each system of the S-SoS to every centroid initialised. Based on the values found, systems are assigned to the centroid with the lowest distance.

There are four types of distances to form clusters to measure as follows.

### Euclidean distance measure

Assuming that there are two systems $p$ and $q$, the Euclidean distance is a standard straight line between them. The calculation is according to Eq. (2).

$$d = \sqrt{\sum_{i=1}^{n} (q_i - p_i)^2} \tag{2}$$

### Squared Euclidean distance measure

It is identical to the Euclidean distance measure but does not take the square root.

$$d = \sum_{i=1}^{n} (q_i - p_i)^2 \tag{3}$$

### Manhattan distance measure

The absolute difference between the two systems sum computed, coordinates are shown in Eq. (4). It is the distance between two systems measured along axes at right angles.

$$d = \sum_{i=1}^{n} |q_x - p_x| + |q_y - p_y| \tag{4}$$

### Cosine distance measure

The angle cosine between the two systems vectors is determined in the space of dimension $n$, as in Eq. (5), considering that $p_i$ and $q_i$ are components of vector $p$ and $q$.

$$d = \frac{\sum_{i=0}^{n-1} q_i - p_i}{\sum_{i=0}^{n-1} q_i^2 \times \sum_{i=0}^{n-1} p_i^2} \tag{5}$$

To form clusters (S-SoSs), first, random points as S-SoS called centroids are assigned. Second, assign each system to the closest S-SoS centroid by implementing a Euclidean distance (the system distance to centroid). Third, identify the new centroids by taking the

average of the assigned systems points, then continue to repeat the second and third steps until convergence is achieved.

It is the first part of the SSBFCSoS algorithm. Conflicts are detected and addressed within each subsystem in the OBFSoS algorithm (*Younes, Ahmed & Elsayed, 2019*), then comes the second part of the clustering method; aggregating all S-SoSs into G-SoS, in stages that mainly depend on the number of S-SoSs.

The last part of the clustering method in SSBFCSoS belongs to the agglomerative hierarchical algorithm, through the bottom-up process. This algorithm is based on finding the distance from the closest pair of S-SoS via the distance between the two centroids $d(S\_SoS_{AB})$. Assuming that the two closest clusters are S-SoS (A) and S-SoS (B), the newly formed cluster would be S-SoS (AB). Reset centroid for the new cluster, and update the distance matrix for these two clusters, giving only a distance between the new S-SoS (AB) and the other remaining clusters, then calculate the average distance between two clusters as the average distance between all systems as in Eq. (6), where $d_{ik}$ represents the distance (similarity). $N(S_{SoSAB})$ and $d(S\_SoS_C)$ are numbers of systems in clusters A, B, and C.

$$d((S\_SoS_{AB}))_C = \frac{\sum_i \sum_k d_{ik}}{N_{(S\_SoS_{AB})} N_{S\_SoS_C}} \tag{6}$$

Finally, detect conflicts inside the new cluster and address them using OBFSoS, then iterate these steps until all clusters merged into a single one called General SoS (G-SoS) so that all systems in G-SoS after the algorithm are terminated.

## PROPOSED METHOD

The proposed method SSBFCSoS considers an enhancement of the OBFSoS method (*Younes, Ahmed & Elsayed, 2019*). Both methods have adopted the concept of better decision making for SoS by addressing conflicts between CS. The proposed SSBFCSoS method consists of three phases, summarised in Fig. 1.

**The first phase** deals with k-means, one of the clustering techniques for dividing CS into groups. Such groups use Java Format, Packages in Glest Game and StarCraft Brood War packages. The requirements are; processor Intel(R) Core (TM) i5-6200U CPU @ 2.30 GHz 2.40 GHz, RAM 8.00 GB and platform is 64-bit Windows 10 operating system, ×64-based processor.

**The second phase** presents the reversal of Glest and StarCraft Brood War code generating into OWL Ontologies of all systems in each cluster, together with merging the OWL ontologies for conflict detection and decision making for each S-SoS. Plus, StarCraft Brood War is converting.

**The third phase** conflict detection of conflicts while merging the S-SoSs using another type of clustering technique; getting G-SoS better decision.

Coordinating systems with common characteristics using Ontology is easier (*Gutierrez, 2018*), where avoiding conflicts resulting from similar systems components. The Belief Function (BF) equation assumes in the OBFSoS algorithm, which used for the SSBFCSoS

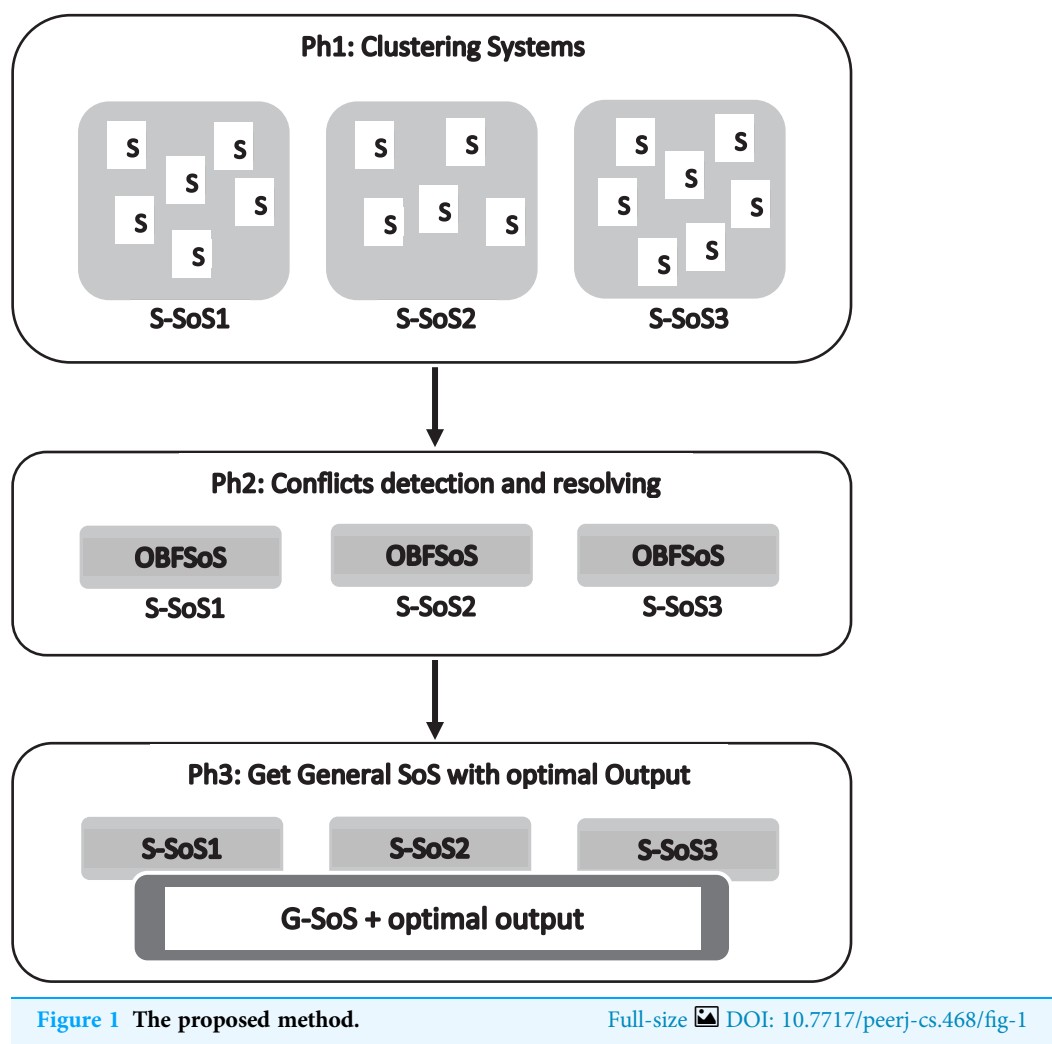

**Figure 1** The proposed method.

method as well. BF primary use is enabling the measurement of conflict weight, and the probability of decision making.

The algorithm is in three parts, two of which are for the clustering processes and the third is for conflict resolution within each cluster summarised in Algorithm 1, which presents the pseudo-code for SSBFCSoS, depending on using two clustering methods; the (k-means) method and the (bottom-up) agglomerative method.

**The first phase** specifies the maximum number of desired systems ($s_i$) in each cluster and the number of desired clusters ($C_l$), then, after determining the number of clusters to be divided for the systems to enter, centroids are were chosen randomly, including the rest of the systems in the clusters, hence, calculating the distance ($d_{s_i}$) for connecting each system in the cluster with the closest centroids.

Calculating the distance according to the location of each system is performed by several methods. Euclidean distance measurement is the adopted method (Eqs. (2) and (3)), since it was more suitable for the homogeneous cases applied, because of the distance calculation based on the raw inputs.

---

**Algorithm 1** Smart Sematic Belief Function Clustered System of Systems (SSBFCSoS).

1. **Input**

   $s_i$  $(s_1, s_2, \ldots, s_n)$  $(i = 2, \ldots, n)$ // Number of desired systems in each cluster

   $C_l$  $(c_1, c_2, \ldots, c_r)$ $(l = 1, 2, \ldots, r)$ // Number of desired clusters

2. $C = 0$ // Cluster Index

3. $ClusteredSystems = 0$

4. *for (each unclustered $s_i$)*

5. $s_i \leftarrow RandomSystems\ (i, n)$ // Choose system at random

6. $C_l \leftarrow s_i$ // Make this random system as first cluster center

7. $C + +$ // initialize new cluster

8. $ClusteredSystems + +$

9. **for** $l \leftarrow$  2 $\ldots r$ **do** // loop over the rest of the centres

10. **for** $i \leftarrow$  1 $\ldots n$ **do** // loop over the systems

11. $d_{s_i} \leftarrow min_{i < 1} \| sqr(d_{s_i} - d_{c_{l_{cent}}}) \|^2$ // Compute the distance to the closest centre

12. *Add $s_i$ to the cluster $C_l$* // Systems enter the closest centroid Clusters

13. **end for**

14. **for** $i \leftarrow$  1 $\ldots n$ **do** // loop over the systems again

15. $c_l \leftarrow d^2{}_n / \sum_i d_i^2$ // Compute a distribution proportional

16. **end for**

17. *Recalculate centroids*

18. $C_r \leftarrow s_n$ // Draw systems to each cluster from distance

19. **end for**

20. *if (ClusteredSystems $=$  $s_n$)*

   *break;* // Clustering is complete

21. *Implement OBFSoS Algorithm between generated clusters results* // $OSoS_x$

22. **for** $l = 1 :$  $r$ **do** // Clusters

23. *Implement OBFSoS for each cluster*

24. **end for**

25. *Get output from OBFSoS $X_{ij}$ , $w_{ij}$* // Decision & Weight

26. $(X_{ij})_{new} \leftarrow X_{ij}$ , $(w_{ij})_{new} \leftarrow w_{ij}$, $(X_{ij})_{old} \leftarrow 0$, $(w_{ij})_{old} \leftarrow 0$

27. **for** $x = 1, 3, 4, \ldots ,$  $r$ **do** // General SoS

28. *if $(w_{ij})_{new} > (w_{ij})_{old}$*

29. $GSoSX_{ij} \leftarrow (X_{ij})_{new}$

30. $(X_{ij})_{old} \leftarrow (X_{ij})_{new}$

31. *else if $(w_{ij})_{old} > (w_{ij})_{new}$*

32. $GSoSX_{ij} \leftarrow (X_{ij})_{old}$

33. *else //* $(w_{ij})_{old} = (w_{ij})_{new}$

34. $GSoSX_{ij} \leftarrow (X_{ij})_{old} \leftarrow (X_{ij})_{new}$

---

| **Algorithm 1** (continued). |
|---|
| 35. $(X_{ij})_{old} \leftarrow (X_{ij})_{new}$ |
| 36. **end for** |
| 37. *A set of n systems for each cluster* |
| 38. **for** $i = 1 : n$ **do** |
| 39. **for** $j = 1 : n$ **do** |
| 40. $d(i,j) = distance\ function\ (c(i), c(j))$ |
| 41. **end for** |
| 42. $pair(i) = \min(d(i, j = 1 : n))$ |
| 43 **end for** |
| 44. **for** $k = 1 : n$ **do** |
| 45. $MS_{SoS} = pair(i)$ |
| 46. *Get min distance* $(MS_{SoS}, S_{SoS})$ |
| 47. $MSoS = merge\ (MS_{SoS} \& S_{SoS})$ |
| 48. *back to step* 13 |
| 49. **end for** |
| 50. $GSoS = MSoS$ |
| **Output** $GSoSX_{ij}$ |

The primary advantage of the Euclidean method is that the distance between any two systems is not affected by adding new systems to the analysis, which may be outliers, however, distances can be significantly affected by differences in scale between the dimensions over which distances calculated. For example, if a dimension denotes the length measured in centimetres, it is preferred not to convert it to another scale, as the resulting Euclidean can be significantly affected. Euclidean distance gives the best result (*Singh, Yadav & Rana, 2013*).

$$J = \sum_{l=1}^{r} \sum_{i=1}^{n} \|s_i^{(l)} - C_l\|^2 \tag{7}$$

Equation (7) calculates the distance for systems, considering $r$ number of clusters, $n$ number of systems $s_i$ case $i$ *and* $C_l$ the centroid for cluster $l$. $J$ is an objective function where, $\|s_i^{(j)} - C_l\|^2$ represent the distance function.

After determining the inputs, all systems are considered part of a cluster (S-SoS), and the selection of the cluster depends on the centroid focal point closest to it. One system in two clusters is not allowed at the same time.

After organising the clusters, each cluster will be addressed as a separate system, so that each cluster will resolve conflicts between the CS, obtaining the best decision and observing all the regulations and tasks related to it. All this is achieved in parallel to the clusters S-SoSs.

**The second phase** is the OBFSoS method role, which primarily depends on addressing conflicts and obtaining the best decision. In this method, weight is relied upon because the system relies on more data for decision-making, this means that it examines more data to make the right decision. The requirement is to obtain a system decision with the most considerable value for weight, suppose the weight values are equal in more than one system. In this case, the solution is either merging the decisions or leaving the decision to the user.

The output variable of the OBFSoS method is $X_{ij}$ for optimising decision making for SoS, the weight variable $w_{ij}$. $X_{ij}$ represents each cluster decision. $w_{ij}$ represents the weight of this decision. Each cluster as a part of General SoS (G-SoS) needs the output maintained while prioritising conflict issues according to SoS requirements, putting into consideration that decision-making precedes delegation decisions.

**The third phase** includes obtaining the information required from each cluster $X_{ij}$ and $w_{ij}$, followed by entering into the final stage of Algorithm 1, to obtain the best Decision regarding G-SoS. This is the stage of comparing the obtained results from the OBFSoS algorithm, comparing the effects and weights of all decisions to enable the G-SoS to obtain the optimal decision.

Another kind of cluster approach has been adopted to finally integrate the S-SoSs into a single SoS (G-SoS). At the end of the algorithm SSBFCSoS, the two closest clusters (S-SoS) are identified by calculating min distance $d(i, j)$, then merging them with the OBFSoS algorithm as in $MS_{SoS}$. Then, although the weights of each S-SoS are preserved, conflicts were detected in the new S-SoSs that merged; a sequence repeated until a G-SoS idealised decision is reached.

There is only one particular case. where there are two groups of equal weights; the two clusters decisions are combined allowing the human element to intervene and reach the right decision.

## CASE STUDY 1 (GLEST)

SSBFCSoS, as an algorithm represented in this paper, applies to many systems states. One of these instances and the experimental setup used for our empirical analysis is described. This case has been selected and set up to match previous work (*Younes, Ahmed & Elsayed, 2019*). Ontology is useful for systems with similar components. Such systems can be called homogeneous systems. One of these homogeneous systems is gaming. The system does not just mean having software, but rather hardware and software, in addition to the human component and the communication between all of these components. This online connexion can be in the case of online games, provided that every player with all these components is considered a system.

The choice was an action-based RTS (Real-Time Strategy) game called Glest (*Dimitriadis, 2009*). Several actions in the game represent data flows, and each group of actions belongs to a category. It is impossible to deal with layers of the same name without conflict. The similarity of names between categories does not necessarily indicate the similarity of the data.

Such similarity and the resulting conflict caused by it have to be avoided, which is the ultimate role of Ontology, depending on the semantic comparison, which differentiates between the classes meanings of the same name and content with a different owner. It is also necessary for SoS to determine the most appropriate decisions. It is for SoS to perform its duties to the fullest.

The systems deal with more than one stage, of which one is the resulting resolution for each cluster S-SoS. The other step is to obtain the final decision resulting from G-SoS. The goal is to determine the winner as a decision, however, the first stage of its decision is the trade-off between winning and action, as in every cluster that plays as an S-SoS has two directions. Suppose each entry system for this S-SoS plays as a separate system, the S-SoS waits for the decision of which systems to be the winner.

Moreover, more than one system within this group played as multiple, here the action differentiated as a decision. Any action is the right one, and the winning decisions are compared to others inside the cluster. The better the number of points paid, the better the decision. The points here represent the data flow; consequently, the validity of the data.

The faster the SoS, the better. So the individual system is handled as an intelligent agent, to save time because intelligence allows the system to avoid its mistakes, repeating them, and repeated conflicts.

## CASE STUDY 2 (STARCRAFT BROOD WAR)

The StarCraft Brood War (*Dor, 2014*) RTS game applied the same rules as Glest. The main focus of this paper will be the final calculations of the conflict and resolution. The conflict of the Brood War represents the differences between players of each cluster of G-SoS, to evaluate the solution selection mechanism for it depending on Algorithm 1.

## RESULTS AND DISCUSSION

### Glest

Both SSBFCSoS and OBFSoS have been implemented. For SSBFCSoS, the systems were divided into clusters. SSBFCSoS is very effective. The two algorithms are based on ontology and BF use. However, SSBFCSoS with this clustering still outperforms OBFSoS in many scenarios.

Each attempt begins with three clusters, and each cluster includes 15–30 systems ranges. The formation of clusters depends on the location of the systems, according to system IP with a time limit of 120 s for both algorithms sets, which is for the OBFSoS algorithm to define conflicts between systems. For SSBFCSoS, the time for each cluster is to identify the systems involved to get conflicts. If an algorithm lacks the ability to resolve an issue within the time limit, it stops, and the failure returned. The goal here is to study the behaviour of the two algorithms for a specific number of systems.

Table 1 shows the number of conflicts generated for SSBFCSoS compared to OBFSoS averaged 70 systems. For ten attempts, in the case of SSBFCSoS, the number of conflicts for the input systems is recorded, with a success rate more than the OBFSoS algorithm.

**Table 1 Glest # of conflicts rate generated for SSBFCSoS vs. OBFSoS.**

| Glest simulation attempts | # of conflicts | |
|---|---|---|
| | SSBFCSoS | OBFSoS |
| 1 | 54 | 131 |
| 2 | 47 | 125 |
| 3 | 45 | 120 |
| 4 | 38 | 115 |
| 5 | 34 | 107 |
| 6 | 30 | 102 |
| 7 | 25 | 95 |
| 8 | 19 | 88 |
| 9 | 17 | 82 |
| 10 | 15 | 70 |

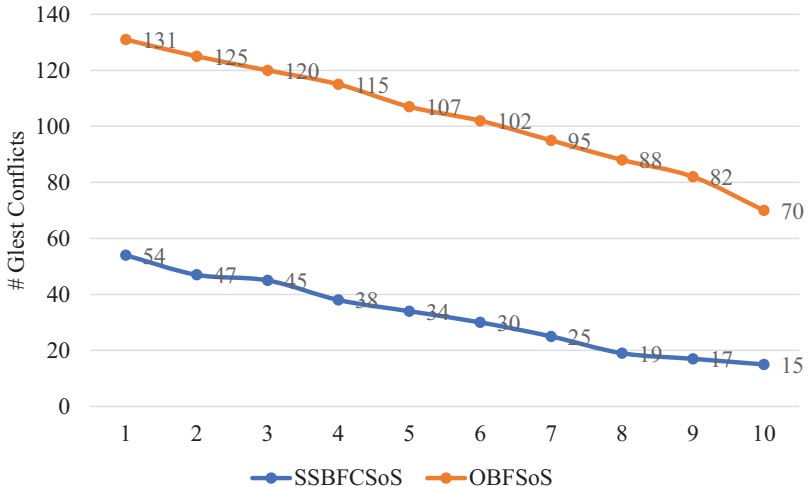

**Figure 2 Conflict rate in Glest for SSBFCSoS vs. OBFSoS.**

Figure 2 shows the conflict rate for SSBFCSoS compared to OBFSoS, which appears for 60 s by the algorithms for several systems. In these minor problems, SSBFCSoS is inferior in conflict rate to OBFSoS, having a slight advantage.

Different algorithms give an instance of a problem with a certain number of $s_i$ systems. The runtime also gives to two algorithms. Each algorithm is given 300 s to resolve the conflicts it encountered. SSBFCSoS divides these systems into S-SoS (clusters) and implements the algorithm associated with each sub-problem separately. Only final results for the SSBFCSoS and OBFSoS algorithms are presented.

There is a sequence in the comparison values between SSBFCSoS and OBFSoS in resolving conflicts regarding decision, design, security, and code. For the first method SSBFCSoS with each cluster value before and after resolving conflicts for all Tables 2, 3, 4,
**Table 2 Glest conflict rate in decisions generated before and after implementing the SSBFCSoS vs. OBFSoS algorithms.**

| Glest simulation attempts | SSBFCSoS | | | | | | | | OBFSoS | |
| --- | --- | --- | --- | --- | --- | --- | --- | --- | --- | --- |
| | Conflicts rate for systems $n = 70$, Run time = 300 s | | | | | | | | | |
| | Before method | | | | After method | | | | #Conflicts before method | #Conflicts after method |
| | Clusters conflicts in decision | | | # Conflicts | Clusters conflicts in decision | | | # Conflicts | | |
| | C1 | C2 | C3 | | C1 | C3 | C3 | | | |
| 1 | 0.250 | 0.350 | 0.400 | 0.370 | 0.200 | 0.300 | 0.300 | 0.296 | 0.397 | 0.366 |
| 2 | 0.250 | 0.375 | 0.375 | 0.340 | 0.188 | 0.313 | 0.375 | 0.298 | 0.408 | 0.360 |
| 3 | 0.267 | 0.400 | 0.333 | 0.333 | 0.133 | 0.267 | 0.267 | 0.222 | 0.417 | 0.350 |
| 4 | 0.333 | 0.333 | 0.333 | 0.316 | 0.167 | 0.250 | 0.250 | 0.211 | 0.417 | 0.339 |
| 5 | 0.273 | 0.273 | 0.455 | 0.324 | 0.091 | 0.182 | 0.364 | 0.206 | 0.411 | 0.346 |
| 6 | 0.222 | 0.222 | 0.556 | 0.300 | 0.111 | 0.111 | 0.333 | 0.133 | 0.422 | 0.343 |
| 7 | 0.500 | 0.167 | 0.333 | 0.240 | 0.333 | 0.167 | 0.167 | 0.160 | 0.421 | 0.326 |
| 8 | 0.250 | 0.250 | 0.500 | 0.211 | 0.250 | 0.000 | 0.250 | 0.105 | 0.398 | 0.341 |
| 9 | 0.667 | 0.000 | 0.333 | 0.176 | 0.333 | 0.000 | 0.000 | 0.059 | 0.415 | 0.341 |
| 10 | 0.333 | 0.333 | 0.333 | 0.200 | 0.000 | 0.000 | 0.333 | 0.067 | 0.414 | 0.357 |

**Table 3 Glest conflict rate in a design generated before and after implementing the SSBFCSoS vs. OBFSoS algorithms.**

| Glest simulation attempts | SSBFCSoS | | | | | | | | OBFSoS | |
| --- | --- | --- | --- | --- | --- | --- | --- | --- | --- | --- |
| | Conflicts rate for systems $n = 70$, Run time = 300 s | | | | | | | | | |
| | Before method | | | | After method | | | | #Conflicts before method | #Conflicts after method |
| | Clusters conflicts in design | | | #Conflicts | Clusters conflicts in design | | | #Conflicts | | |
| | C1 | C2 | C3 | | C1 | C2 | C3 | | | |
| 1 | 0.308 | 0.231 | 0.462 | 0.241 | 0.231 | 0.154 | 0.308 | 0.167 | 0.282 | 0.260 |
| 2 | 0.273 | 0.273 | 0.455 | 0.234 | 0.182 | 0.182 | 0.364 | 0.170 | 0.280 | 0.264 |
| 3 | 0.250 | 0.417 | 0.333 | 0.267 | 0.167 | 0.250 | 0.250 | 0.178 | 0.275 | 0.250 |
| 4 | 0.300 | 0.400 | 0.300 | 0.263 | 0.200 | 0.200 | 0.200 | 0.158 | 0.278 | 0.243 |
| 5 | 0.222 | 0.444 | 0.333 | 0.265 | 0.111 | 0.333 | 0.111 | 0.147 | 0.290 | 0.252 |
| 6 | 0.375 | 0.375 | 0.250 | 0.267 | 0.375 | 0.125 | 0.125 | 0.167 | 0.284 | 0.235 |
| 7 | 0.286 | 0.571 | 0.143 | 0.280 | 0.143 | 0.286 | 0.000 | 0.120 | 0.305 | 0.242 |
| 8 | 0.000 | 0.333 | 0.667 | 0.316 | 0.000 | 0.167 | 0.167 | 0.105 | 0.295 | 0.250 |
| 9 | 0.400 | 0.600 | 0.000 | 0.294 | 0.200 | 0.200 | 0.000 | 0.118 | 0.293 | 0.244 |
| 10 | 0.250 | 0.500 | 0.250 | 0.267 | 0.000 | 0.250 | 0.000 | 0.067 | 0.286 | 0.243 |

and 5, in contrast to the other method OBFSoS in which the conflicts and the resolution for the whole SoS are without clustering.

The decision is to determine which system got the best points to win the game. That is with regards to systems such as an RTS game as the Glest, as a result of making the right decisions. As part of the algorithm, BF is used for the validity of action sources for

**Table 4 Glest conflict rate in code generated before and after implementing the SSBFCSoS vs. OBFSoS algorithms.**

| Glest simulation attempts | SSBFCSoS | | | | | | | | OBFSoS | |
|---|---|---|---|---|---|---|---|---|---|---|
| | Conflicts rate for systems $n$ = 70, Run time = 300 s | | | | | | | | | |
| | Before method | | | | After method | | | | #Conflicts before method | #Conflicts after method |
| | Clusters conflicts in code | | | #Conflicts | Clusters conflicts in code | | | #Conflicts | | |
| | C1 | C2 | C3 | | C1 | C2 | C3 | | | |
| 1 | 0.417 | 0.250 | 0.333 | 0.222 | 0.250 | 0.167 | 0.167 | 0.130 | 0.153 | 0.145 |
| 2 | 0.333 | 0.333 | 0.333 | 0.255 | 0.083 | 0.167 | 0.167 | 0.106 | 0.152 | 0.136 |
| 3 | 0.455 | 0.273 | 0.273 | 0.244 | 0.182 | 0.091 | 0.091 | 0.089 | 0.150 | 0.125 |
| 4 | 0.300 | 0.300 | 0.400 | 0.263 | 0.100 | 0.100 | 0.200 | 0.105 | 0.139 | 0.122 |
| 5 | 0.333 | 0.222 | 0.444 | 0.265 | 0.111 | 0.111 | 0.111 | 0.088 | 0.140 | 0.121 |
| 6 | 0.250 | 0.375 | 0.375 | 0.267 | 0.125 | 0.125 | 0.125 | 0.100 | 0.137 | 0.127 |
| 7 | 0.375 | 0.250 | 0.375 | 0.320 | 0.125 | 0.000 | 0.125 | 0.080 | 0.126 | 0.116 |
| 8 | 0.167 | 0.667 | 0.167 | 0.316 | 0.000 | 0.167 | 0.000 | 0.053 | 0.148 | 0.125 |
| 9 | 0.400 | 0.600 | 0.000 | 0.294 | 0.000 | 0.200 | 0.000 | 0.059 | 0.146 | 0.110 |
| 10 | 0.000 | 0.400 | 0.600 | 0.333 | 0.000 | 0.000 | 0.200 | 0.067 | 0.157 | 0.114 |

**Table 5 Glest conflict rate in security generated before and after implementing the SSBFCSoS vs. OBFSoS algorithms.**

| Glest simulation attempts | SSBFCSoS | | | | | | | | OBFSoS | |
|---|---|---|---|---|---|---|---|---|---|---|
| | Conflicts rate for systems $n$ = 70, Run time = 300 s | | | | | | | | | |
| | Before method | | | | After method | | | | #Conflicts before method | #Conflicts after method |
| | Clusters conflicts in security | | | #Conflicts | Clusters conflicts in security | | | #Conflicts | | |
| | C1 | C2 | C3 | | C1 | C2 | C3 | | | |
| 1 | 0.444 | 0.222 | 0.333 | 0.167 | 0.333 | 0.111 | 0.222 | 0.111 | 0.168 | 0.153 |
| 2 | 0.375 | 0.375 | 0.250 | 0.170 | 0.250 | 0.250 | 0.125 | 0.106 | 0.160 | 0.152 |
| 3 | 0.286 | 0.286 | 0.429 | 0.156 | 0.143 | 0.143 | 0.286 | 0.089 | 0.158 | 0.142 |
| 4 | 0.500 | 0.500 | 0.000 | 0.158 | 0.333 | 0.333 | 0.000 | 0.105 | 0.165 | 0.130 |
| 5 | 0.400 | 0.200 | 0.400 | 0.147 | 0.200 | 0.200 | 0.200 | 0.088 | 0.159 | 0.131 |
| 6 | 0.200 | 0.400 | 0.400 | 0.167 | 0.200 | 0.200 | 0.000 | 0.067 | 0.157 | 0.127 |
| 7 | 0.750 | 0.250 | 0.000 | 0.160 | 0.250 | 0.250 | 0.000 | 0.080 | 0.147 | 0.126 |
| 8 | 0.000 | 0.667 | 0.333 | 0.158 | 0.000 | 0.333 | 0.000 | 0.053 | 0.159 | 0.125 |
| 9 | 0.500 | 0.250 | 0.250 | 0.235 | 0.250 | 0.000 | 0.000 | 0.059 | 0.146 | 0.122 |
| 10 | 0.667 | 0.000 | 0.333 | 0.200 | 0.000 | 0.000 | 0.000 | 0.067 | 0.143 | 0.114 |

each system (player), and using a fewer number of right actions with the same result giving the player the right to win the opponent, even if their points are equal, thus, the decision to win.

Table 2 shows the rate of conflicts in the decisions and their solutions between SSBFCSoS and OBFSoS. The number of attempts denotes the simulation attempts column.

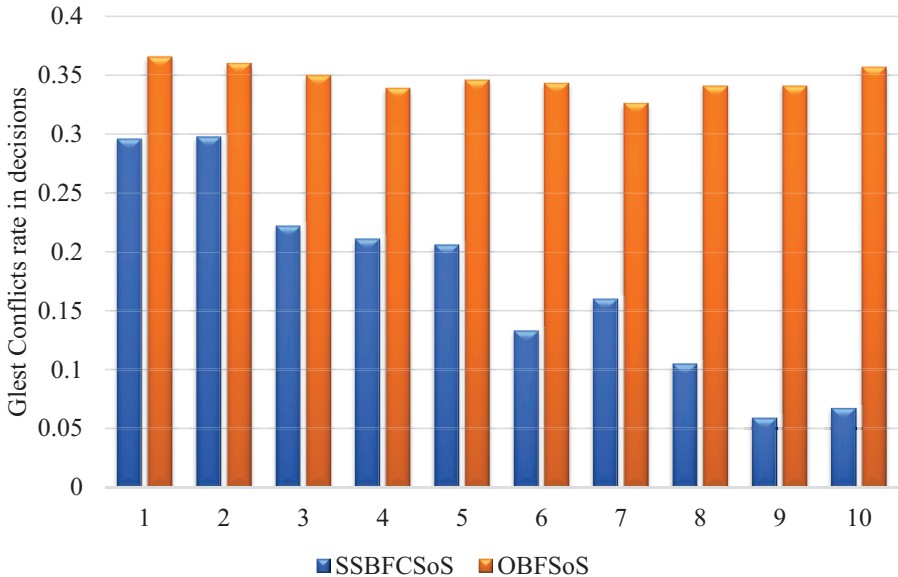

**Figure 3  Glest conflict rate in decisions after implementing SSBFCSoS vs. OBFSoS.**

For SSBFCSoS, the number of desired clusters is indicated by $C_l$. For the given number of $s_i$ systems (70), the best-performing algorithm result is given in bold as in SSBFCSoS.

In Table 2, the values that resulted from applying clustering through SSBFCSoS has shown better results. Attempt 5, for instance, shows the decrease of the conflicts percentage to 0.206 for all the clusters in SSBFCSoS compared to 0.346 for the other method. The results have progressed to reach 0.067 for SSBFCSoS compared to 0.357 for OBFSoS for the last attempt.

Figure 3 illustrates the rate of remaining conflicts in decisions while resolving the SSBFCSoS and OBFSoS algorithms. It illustrates how SSBFCSoS encounters more conflicts on time, and is, therefore, more suitable for SoS than OBFSoS.

Compared to OBFSoS, the SSBFCSoS algorithm curve shows that the rate of conflicts decreases sequentially, as shown in Fig. 3, due to the clustering of conflict handling. The system is an intelligent, learned system that avoids the same conflicts encountered in earlier attempts. It appears, as shown in Table 1, that applying the algorithm SSBFCSoS, the total number of conflicts decreases as the number of attempts increases, so the relationship here is considered inverse.

The system design is the general structure of the system. The parts that this system consists of include units, interfaces, and data. As a system, the game also has a pattern or design. Furthermore, from it, the incompatibility between the structure of two systems (two games) or more, is what calls for design conflicts. Although systems are homogenous, different versions of the game can cause such design conflicts. The same thing may cause code conflicts, as well.

The goal is to make the right decision to determine the winning player and also collect points. In case there are design conflicts between two or more systems, the right decision cannot be made, because of the incompatibility between the design of the two systems.

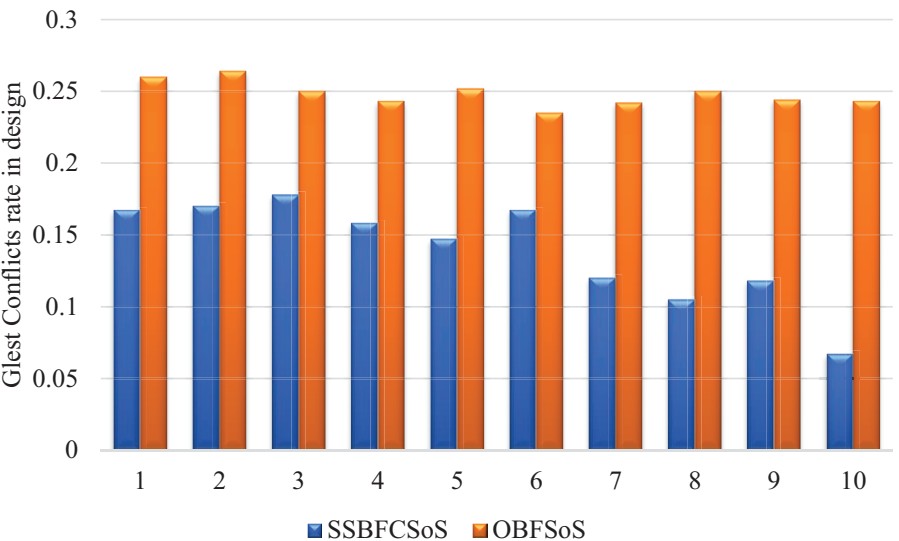

**Figure 4  Glest conflict rate in design after implementing SSBFCSoS vs. OBFSoS.**

According to the rules, to determine the best work, the environment should be equal, since incompatible environments lead to design conflicts, depending on the different versions of the game. If Algorithm is used, SSBFCSoS determines the compatibility. BF also returns to the game source to verify this.

Table 3 displays the rate of conflicts in design with its solutions between SSBFCSoS and OBFSoS. For example, in the last attempt, conflicts in the design percentage reached a ratio of 0.067 in SSBFCSoS compared to 0.243 in OBFSoS, which is considered a good result, as shown in Table 3 that almost all the conflicts in two clusters were dealt with against one conflict in the last cluster concerning method SSBFCSoS. While OBFSoS needed a longer time to reach the same results. The clustering process dealt with conflicts in parallel, thus it consumed less time than without it.

Figure 4 illustrates the rate of remaining conflicts in design after they encountered, during the process of resolving the SSBFCSoS and OBFSoS algorithms. It describes how SSBFCSoS faces more conflicts than OBFSoS in a limited time.

In Table 4, the results of the comparison of SSBFCSoS and OBFSoS shown in the code conflicts. In addition to clarifying the clusters, each cluster is presented separately before and after applying the SSBFCSoS algorithm. SSBFCSoS was able to face conflicts almost double the other method. Whereas for SSBFCSoS, the conflict ratio has decreased to about 0.067 compared to 0.114 for OBFSoS. Either way, it consumed about the same time and could be less.

Figure 5 shows the sequence of results for both algorithms to resolve conflicts in code. The higher the number of attempts, the better the outcome for each algorithm, however, algorithm SSBFCSoS still produces significantly better results as in the figure.

In regards to priority for each system, it may be a security priority. Some systems place security at the top of their interests, according to their goals. In this paper, the systems are gaming systems. Moreover, the game has an online version, therefore, security

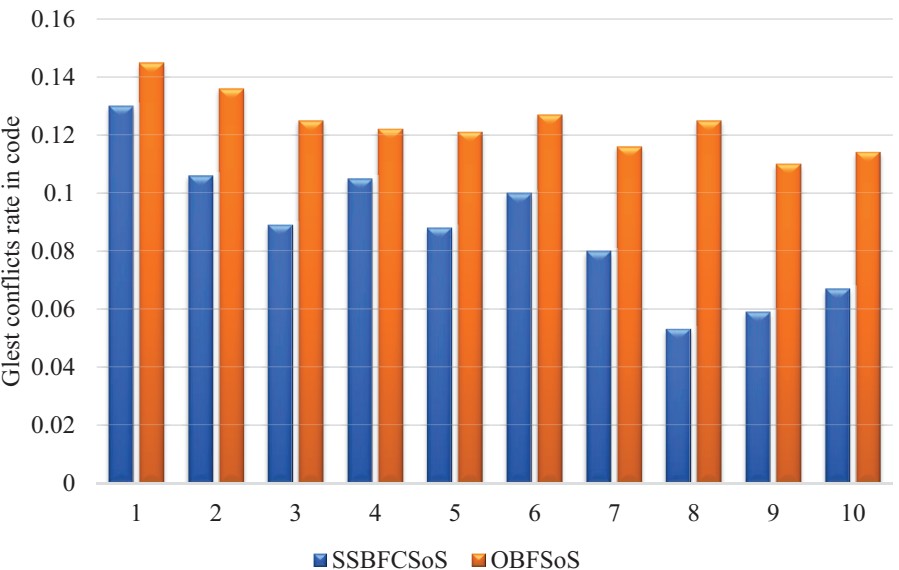

**Figure 5 Glest conflict rate in code after implementing SSBFCSoS vs. OBFSoS.**

becomes essential. The importance of security lies in its influence on the decision of our SoS. So, it is considered working for security if it affects the decision because getting a better decision is the first and primary goal.

The score has to be confirmed to determine who won, which is why, in this case, not all players can enter the game until they pass the registration rules. That is to achieve the security requirement. For that, Table 5 records the evolution in the number of conflicts in security. In Table 5, SSBFCSoS achieved good results in facing security conflicts as much as possible. The conflicts ratio was at attempt 7 was about 0.080 and 0.126 for OBFSoS for the same attempt.

Figure 6 shows the rate of development of security conflicts after applying methods, showing satisfactory results by the SSBFCSoS in comparison to OBFSoS.

Table 6 shows the conflict rate resolved by the algorithms, and the time takes for each algorithm to obtain these solutions. We have reached a better SSBFCSoS algorithm to get good results, which consumes a shorter time. As shown in Table 6, the first attempt, the resolving conflicts percentage for the SSBFCSoS has reached 0.296 within approximately 281 s, while OBFSoS achieved 0.076 within 300 s. As for the last attempt, the rate of resolving conflicts has reached 0.733 for the SSBFCSoS within 113 s, while OBFSoS has achieved about 0.171 within 0.276 s.

These rates show in Fig. 7, which reflects the rate of the conflicts resolved.

Figure 8 shows a comparison of the time taken for each algorithm to obtain solutions, proving that the SSBFCSoS algorithm is the best in terms of time consumption.

## STARCRAFT BROOD WAR

To evaluate the StarCraft Brood War, the conflict resolution rates in the cases under consideration generated by SSBFCSoS and OBFSoS are calculated and listed in Table 7.

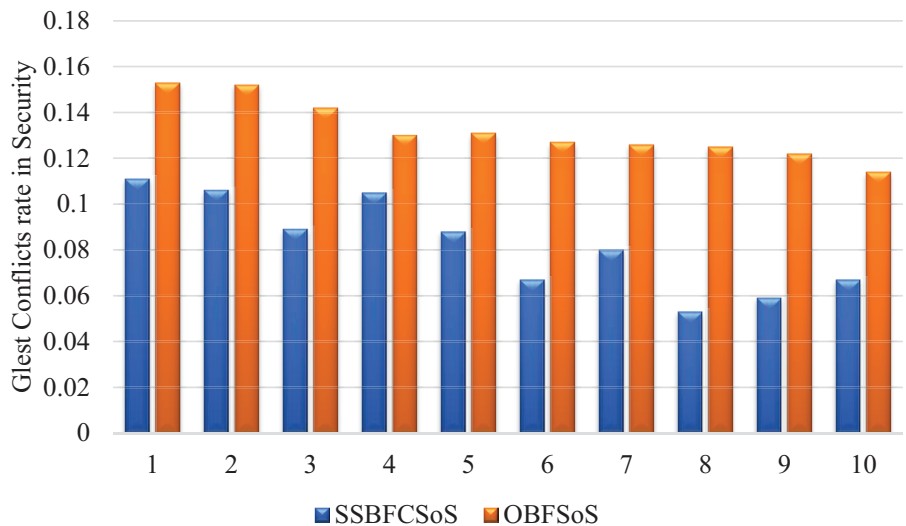

**Figure 6 Glest conflict rate in security after implementing SSBFCSoS vs. OBFSoS.**

**Table 6 Glest # of resolved conflicts rate generated SSBFCSoS vs. OBFSoS with consumed time.**

| Glest simulation attempts | SSBFCSoS | | OBFSoS | |
|---|---|---|---|---|
| | Time | Rate | Time | Rate |
| 1 | 0.296 | 281.11 | 0.076 | 300.00 |
| 2 | 0.319 | 270.59 | 0.088 | 300.00 |
| 3 | 0.422 | 262.28 | 0.133 | 288.89 |
| 4 | 0.421 | 214.67 | 0.165 | 278.26 |
| 5 | 0.471 | 204.44 | 0.15 | 283.49 |
| 6 | 0.533 | 180.39 | 0.167 | 277.78 |
| 7 | 0.560 | 161.40 | 0.189 | 270.18 |
| 8 | 0.684 | 161.85 | 0.159 | 280.30 |
| 9 | 0.706 | 122.34 | 0.183 | 272.36 |
| 10 | 0.733 | 113.58 | 0.171 | 276.19 |

For decision conflicts appearance as in attempt 9, the SSBFCSoS has reached 0.250 compared to 0.394 by OBFSoS. For design conflicts appearance as in attempt 8, the SSBFCSoS has reached 0.250 compared to 0.287 by OBFSoS. As for security conflicts in the same attempt, SSBFCSoS has reached 0.167 compared to 0.204 by OBFSoS.

Table 8, shows that SSBFCSoS has achieved superiority over OBFSoS in terms of conflict resolution in SoS.

Table 8 presents the number of conflicts generated for SSBFCSoS compared to OBFSoS with an average of 100 systems for G-SoS. SSBFCSoS divides these systems into 3 clusters (S-SoSs), with the number of systems ranging from 25 to 36 per cluster. Each algorithm is allowed 400 se to resolve the conflicts encountered.

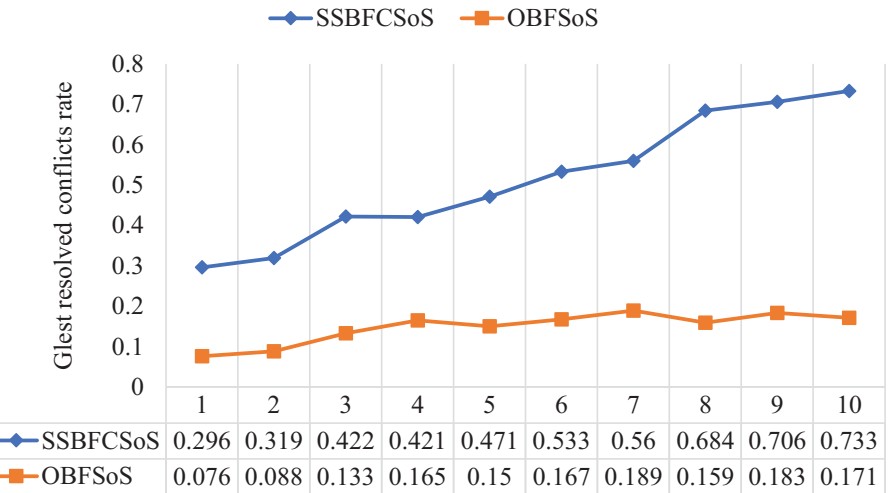

| | 1 | 2 | 3 | 4 | 5 | 6 | 7 | 8 | 9 | 10 |
|---|---|---|---|---|---|---|---|---|---|---|
| SSBFCSoS | 0.296 | 0.319 | 0.422 | 0.421 | 0.471 | 0.533 | 0.56 | 0.684 | 0.706 | 0.733 |
| OBFSoS | 0.076 | 0.088 | 0.133 | 0.165 | 0.15 | 0.167 | 0.189 | 0.159 | 0.183 | 0.171 |

**Figure 7 Glest resolved conflict rate after implementing SSBFCSoS vs. OBFSoS.**

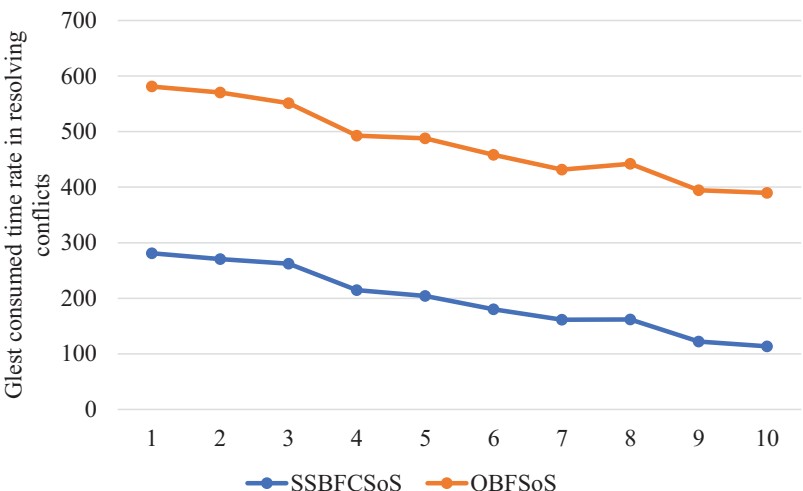

**Figure 8 Glest Time Consumed in Conflicts Resolving during implementing SSBFCSoS vs. OBFSoS.**

Table 8 shows only the final result for the three merged clusters in decision, design, code and security conflicts. At attempt 8, the percentage of conflicts after applying SSBFCSoS has reached 0.208 compared to 0.389 for OBFSoS in decision conflicts. In design, conflicts have reached 0.185 for SSBFCSoS compared to 0.242 for OBFSoS in attempt 7. In attempt 9, the SSBFCSoS conflict ratio has reached 0.100 compared to 0.154 for OBFSoS in terms of security, and the code conflicts ratio has reached 0.100 for SSBFCSoS compared to 0.163 for OBFSoS.

Figure 9 shows the improvement in the conflicts rate in decisions for SoS achieved by SSBFCSoS in comparison to that achieved by OBFSoS.

Figure 10 shows that the conflicts in the design reduced by using SSBFCSoS better than OBFSoS.

**Table 7 StarCraft Brood War Conflicts rate generated for SSBFCSoS vs. OBFSoS.**

| StarCraft Brood War simulation attempts | SSBFCSoS | | | | OBFSoS | | | |
|---|---|---|---|---|---|---|---|---|
| | Conflicts rate for systems $n = 100$, Run time = 400 s | | | | | | | |
| | Conflicts ratio | | | | Conflicts ratio | | | |
| | Decision | Design | Security | Code | Decision | Design | Security | Code |
| 1 | 0.343 | 0.239 | 0.209 | 0.209 | 0.364 | 0.283 | 0.192 | 0.162 |
| 2 | 0.322 | 0.254 | 0.203 | 0.220 | 0.385 | 0.275 | 0.181 | 0.159 |
| 3 | 0.314 | 0.255 | 0.216 | 0.216 | 0.390 | 0.273 | 0.174 | 0.163 |
| 4 | 0.319 | 0.255 | 0.191 | 0.234 | 0.383 | 0.272 | 0.185 | 0.160 |
| 5 | 0.317 | 0.268 | 0.195 | 0.220 | 0.396 | 0.268 | 0.181 | 0.154 |
| 6 | 0.303 | 0.242 | 0.212 | 0.242 | 0.396 | 0.273 | 0.180 | 0.151 |
| 7 | 0.333 | 0.222 | 0.222 | 0.222 | 0.398 | 0.273 | 0.172 | 0.156 |
| 8 | 0.333 | 0.250 | 0.167 | 0.250 | 0.435 | 0.287 | 0.204 | 0.074 |
| 9 | 0.250 | 0.300 | 0.200 | 0.250 | 0.394 | 0.279 | 0.173 | 0.154 |
| 10 | 0.278 | 0.278 | 0.222 | 0.222 | 0.402 | 0.278 | 0.186 | 0.134 |

**Table 8 StarCraft Brood War Resolved Conflicts generated after implementing the SSBFCSoS vs. OBFSoS.**

| StarCraft Brood War simulation attempts | SSBFCSoS | | | | OBFSoS | | | |
|---|---|---|---|---|---|---|---|---|
| | Conflicts rate for systems $n = 100$, Run time = 400 s | | | | | | | |
| | Resolved conflicts ratio | | | | Resolved conflicts ratio | | | |
| | Decision | Design | Security | Code | Decision | Design | Security | Code |
| 1 | 0.269 | 0.224 | 0.164 | 0.149 | 0.343 | 0.258 | 0.177 | 0.157 |
| 2 | 0.254 | 0.220 | 0.169 | 0.153 | 0.352 | 0.264 | 0.181 | 0.159 |
| 3 | 0.255 | 0.235 | 0.157 | 0.157 | 0.355 | 0.250 | 0.163 | 0.163 |
| 4 | 0.255 | 0.234 | 0.149 | 0.149 | 0.352 | 0.241 | 0.167 | 0.167 |
| 5 | 0.268 | 0.195 | 0.146 | 0.122 | 0.362 | 0.248 | 0.161 | 0.161 |
| 6 | 0.212 | 0.182 | 0.152 | 0.121 | 0.367 | 0.245 | 0.158 | 0.158 |
| 7 | 0.222 | 0.185 | 0.148 | 0.111 | 0.375 | 0.242 | 0.156 | 0.156 |
| 8 | 0.208 | 0.208 | 0.125 | 0.083 | 0.389 | 0.259 | 0.176 | 0.167 |
| 9 | 0.100 | 0.200 | 0.100 | 0.100 | 0.365 | 0.240 | 0.154 | 0.163 |
| 10 | 0.056 | 0.056 | 0.111 | 0.111 | 0.309 | 0.227 | 0.155 | 0.144 |

Figure 11 with almost the same conflict rate for SSBFCSoS and OBFSoS. However, SSBFCSoS can resolve more conflicts than OBFSoS.

The curve in Fig. 12 illustrates the code conflicts of the two methods.

Table 9 shows the rate of addressing conflicts of the SoS in the ratio of both SSBFCSoS and OBFSoS along with the time taken. In attempt 7, the percentage of solutions achieved by SSBFCSoS has reached 0.333 within 127 s compared to a ratio of 0.70 within 317 seconds for OBFSoS.

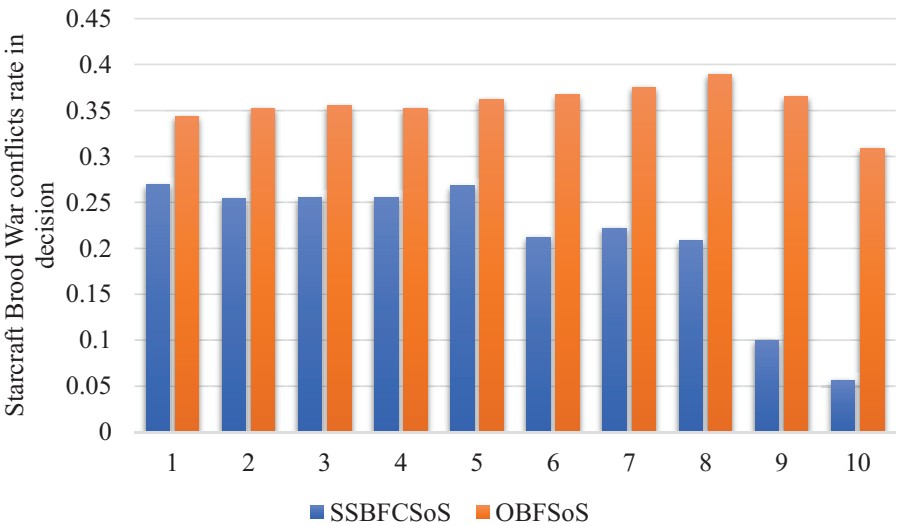

**Figure 9  StarCraft Brood War Conflicts rate in decisions after implementing SSBFCSoS vs. OBFSoS.**

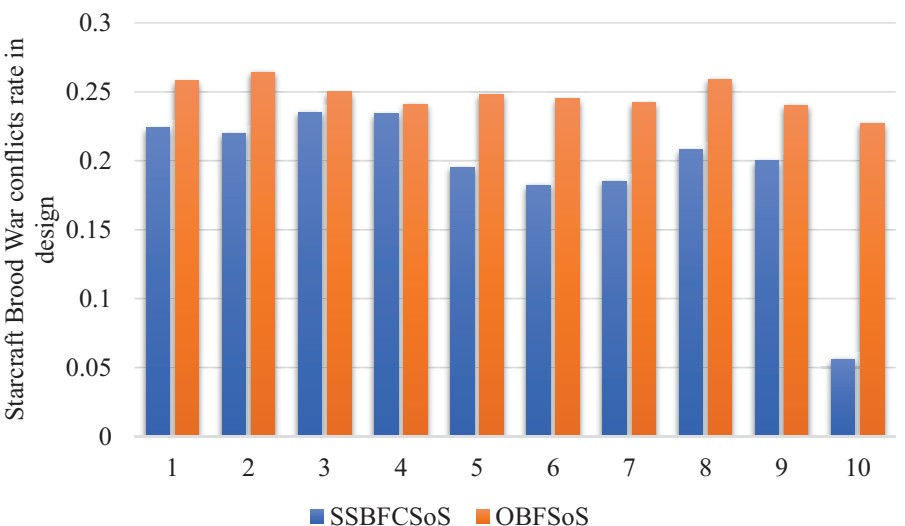

**Figure 10  StarCraft Brood War Conflicts rate in design after implementing SSBFCSoS vs. OBFSoS.**

Figure 13 shows the extent to which SSBFCSoS was better than OBFSoS in terms of addressing conflicts, and its rate has reached 67% in attempt 10 compared to 17% for OBFSoS.

Figure 14 Compares the time spent for each method, showing the advantage of SSBFCSoS in saving time at a rate of more than three times the time consumed by OBFSoS.

Each of the two systems revealed the number of conflicts. The results indicated the progress of SSBFCSoS, which has reached this success thanks to the division of inputs from the systems to fall under the name clusters S-SoS. It also shows an apparent effect on

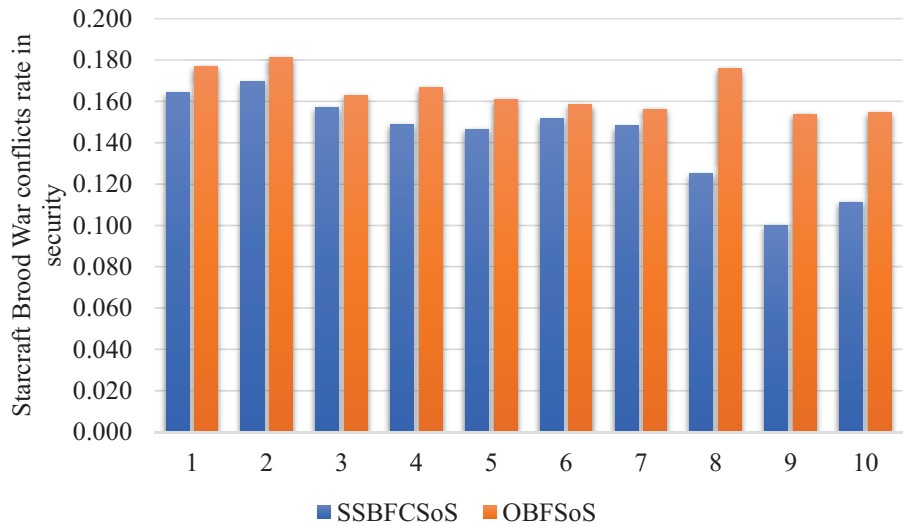

**Figure 11  StarCraft Brood War Conflicts rate in security after implementing SSBFCSoS vs. OBFSoS.**

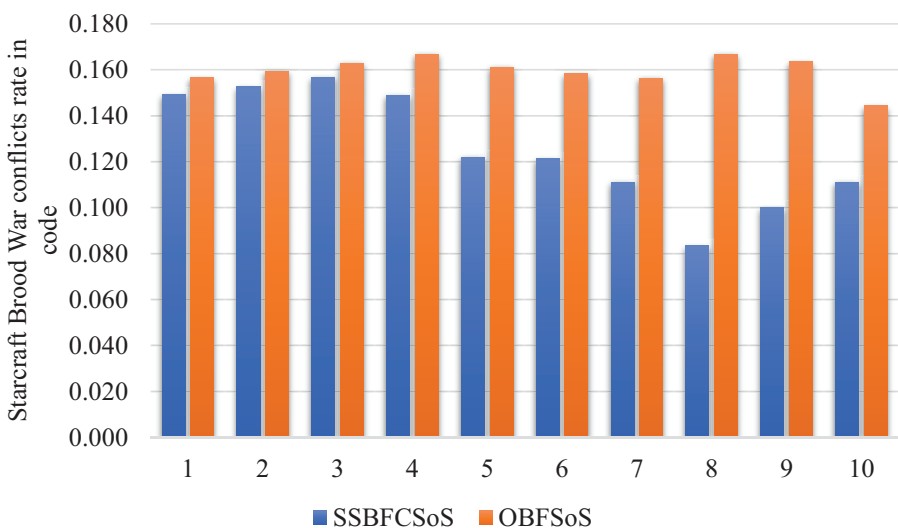

**Figure 12  StarCraft Brood War Conflicts rate in code after implementing SSBFCSoS vs. OBFSoS.**

the proper use of time. Focusing on time is essential in exceptional cases, such as the systems to which it was applied.

## Comparing the two methods SSBFCSoS and OBFSoS

Experience clearly shows that there is no established law for making a better Decision for SoS. However, the performance of each of the applied algorithms depends on the features of the conflicts.

At the beginning of the application, a certain number—three—of clusters are identified, with each cluster containing 20 to 35 systems, ultimately reaching about 70 or 100 systems. Time is set to approximately 120 s to identify conflicts, and about 300 s to address

**Table 9 StarCraft Brood War resolved conflict rate generated after implementing the SSBFCSoS vs. OBFSoS with consumed time.**

| StarCraft Brood War simulation attempts | SSBFCSoS | | OBFSoS | |
|---|---|---|---|---|
| | Rate | Time | Rate | Time |
| 1 | 0.194 | 369.15 | 0.066 | 399.28 |
| 2 | 0.203 | 251.26 | 0.044 | 377.43 |
| 3 | 0.196 | 198.5 | 0.070 | 370.44 |
| 4 | 0.213 | 189.34 | 0.074 | 355.31 |
| 5 | 0.268 | 158.55 | 0.067 | 341.37 |
| 6 | 0.333 | 131.51 | 0.072 | 335.58 |
| 7 | 0.333 | 127.28 | 0.070 | 317.08 |
| 8 | 0.375 | 101.45 | 0.009 | 304.21 |
| 9 | 0.500 | 98.56 | 0.077 | 299.25 |
| 10 | 0.667 | 87.54 | 0.165 | 289.11 |

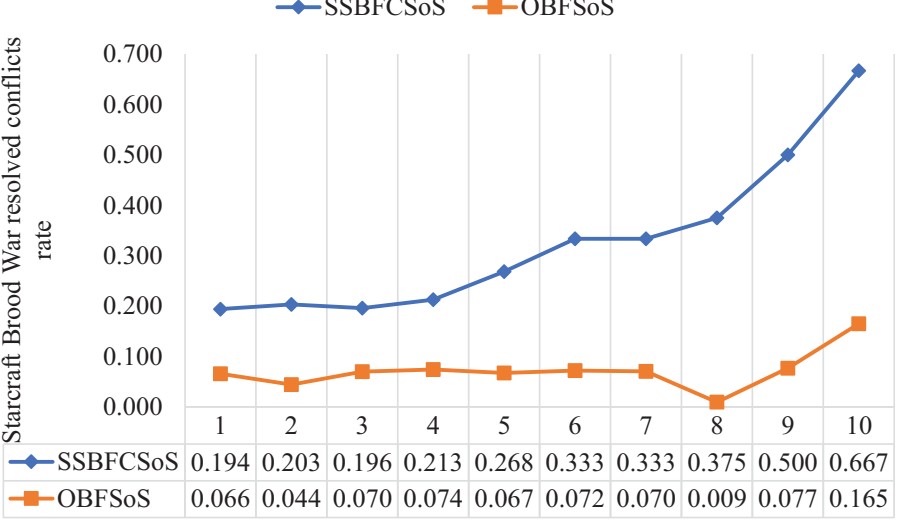

**Figure 13 StarCraft Brood War resolved conflict rate after implementing SSBFCSoS vs. OBFSoS.**

those conflicts. These features can be of conflict type and number of systems that have conflict at the same time, in addition to the number of conflicts encountered during the solution process.

Each algorithm has its method depending on the case. Nevertheless, through research, a method of evaluating the performance of each algorithm has been proposed. So, we present the following general trends that we observed:

- The algorithm SSBFCSoS outperforms the other algorithm in most cases with an unlimited number of systems.
- The longer the time is taken by algorithm OBFSoS, the better-achieved results, but to a limited number of systems only.

- As for the clusters in algorithm SSBFCSoS, the system must not be part of more than one cluster simultaneously.
- The game running online, allows the system to exit and enter more than once. It could create a repeated conflict, especially in security.
- Although the algorithm system SSBFCSoS is a learned system. However, it can seriously increase in some attempts over others in conflicts for the previously mentioned reasons.
- Continuous attempts with algorithm SSBFCSoS reduce the number of conflicts and the time taken to resolve conflicts.
- There is a constant conflict despite repeated attempts, which requires user intervention to get the appropriate and better decision. Therefore, a significant percentage of about 20% of the unresolved conflicts was required for user intervention.

Figure 15 shows the number of systems and clusters and the conflict resolution rate. The number of clusters was set proportional to the number of systems, creating a direct relationship; the more systems and clusters, the more satisfactory the results. Should we let the system limit the number of clusters? If we leave it open, it is possible to create semi-empty clusters containing only two or three systems. So, we tried to define the systems. During implementation, we found that if the average number of systems within each cluster was 10% of the total number of systems, the results were very appropriate, and the time was right. Also, if the average systems for each cluster are 25–30, the results are perfect.

Also, we found a second situation during implementation where two or more systems (players) play with the multiplayer system, as in the two systems produce one combined decision. First, we need to compare procedures by OBFSoS algorithm, then compare their decisions. We have two parts to compare the procedures for obtaining the highest points first and determining the best procedure for obtaining higher points for their group, then compare the decisions to get an ideal decision. And then they entered as part of the S-SoS and so on.

## Comparing clustering methods k-means and agglomerative (bottom-up)

We made a statistic that compared two clustering algorithms, namely k-means and agglomerative (bottom-up). The two algorithms were compared in the second case study; StarCraft Brood War. We made this statistic on different numbers of systems, as shown in Table 10. We were beginning with fewer systems, ending with more. We recorded the values after seven attempts. Concerning the algorithm k-means, we specified the number of clusters as 5 clusters. Contrary to k-means, the number of clusters in agglomerative (bottom-up) is not determined, but each system is considered as a separate cluster until several clusters formed. The same distance measurement is used for the k-means.

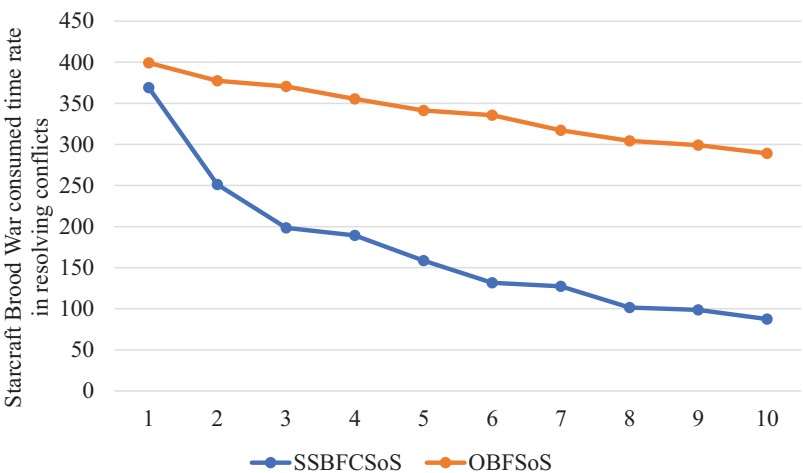

**Figure 14 StarCraft Brood War Time consumed in conflicts resolving during implementing SSBFCSoS vs. OBFSoS.**

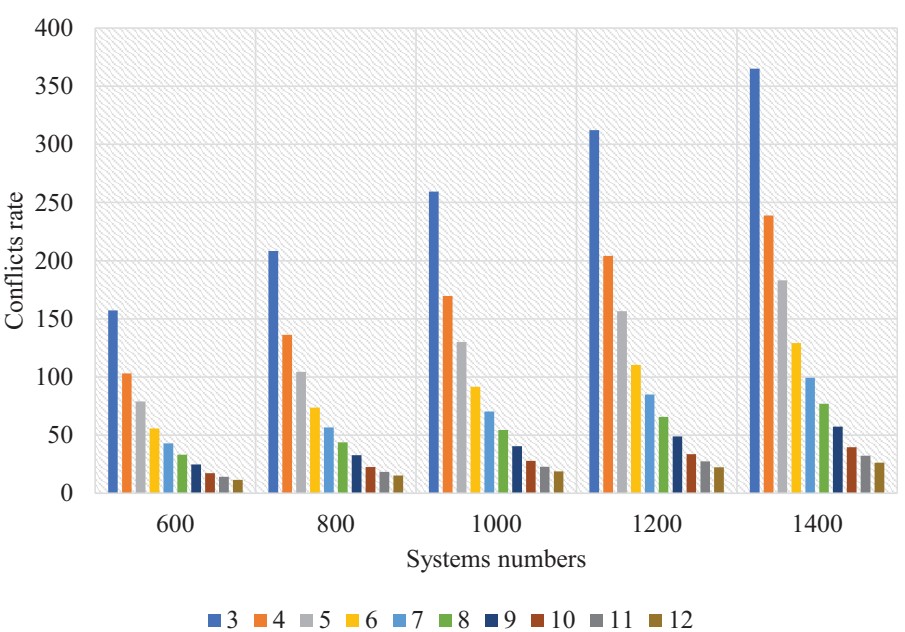

**Figure 15 Resolved conflicts rate for SSBFCSoS with different cluster numbers and systems.**

In Table 10 for case 7 with 110 systems as inputs, k-means achieved a conflict resolution rate of about 90% compared to about 69% for the agglomerative (bottom-up). These results took about 279 seconds for the k-means compared to 790 s for the other one.

Figure 16 clarifies the comparison curve between the two-clustering methods k-means and agglomerative (bottom-up) so that k-means showed excellent results better than the other method.

**Table 10 StarCraft Brood War resolving conflicts rate generated after implementing the SSBFCSoS method using k-means clustering vs. Agglomerative (bottom-up) with consumed time.**

| # systems | k-means | | Agglomerative (bottom-up) | |
|---|---|---|---|---|
| | Rate | Time | Rate | Time |
| 20 | 0.846 | 60.42 | 0.692 | 99.52 |
| 35 | 0.778 | 99.38 | 0.704 | 161.55 |
| 65 | 0.895 | 121.05 | 0.667 | 232.32 |
| 80 | 0.855 | 196.57 | 0.696 | 333.33 |
| 95 | 0.884 | 241.47 | 0.663 | 423.45 |
| 110 | 0.899 | 279.42 | 0.687 | 789.50 |
| 130 | 0.888 | 329.36 | 0.664 | 953.36 |
| 140 | 0.876 | 373.16 | 0.664 | 1,696.17 |
| 170 | 0.884 | 406.38 | 0.661 | 1,900.53 |
| 190 | 0.891 | 539.47 | 0.652 | 2,884.42 |

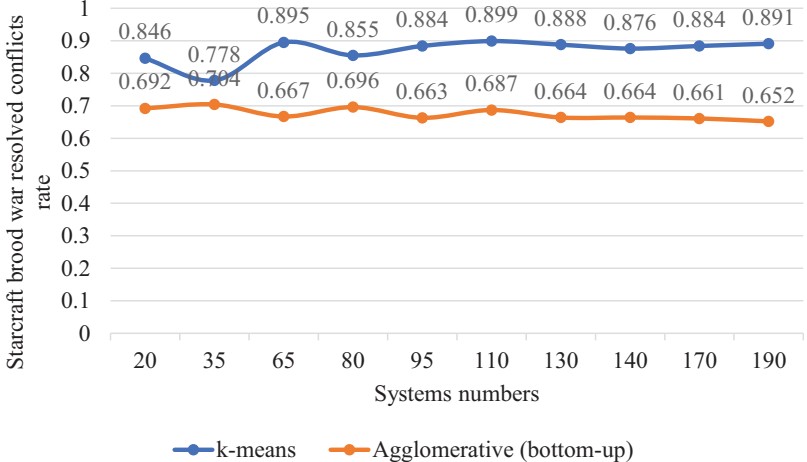

**Figure 16 StarCraft Brood War resolved conflicts rate generated implementing the SSBFCSoS using k-means clustering vs. Agglomerative (bottom-up).**

Figure 17 gave a curve for the time consumed for each method to obtain these results, showing a significant difference in the time taken between the two methods, with k-means saving time while preserving its results better than the other method.

Generally, the k-means algorithm is easy to implement and more suitable with big data such as SoS (*Nasraoui & N'Cir, 2019*).

## Comparison between the k-means algorithm and some clustering algorithms

The choice of the k-mean algorithm as a clustering technique for optimising SSBFCSoS is supported. SoS features are a combination of some IoT features and some of the big data,

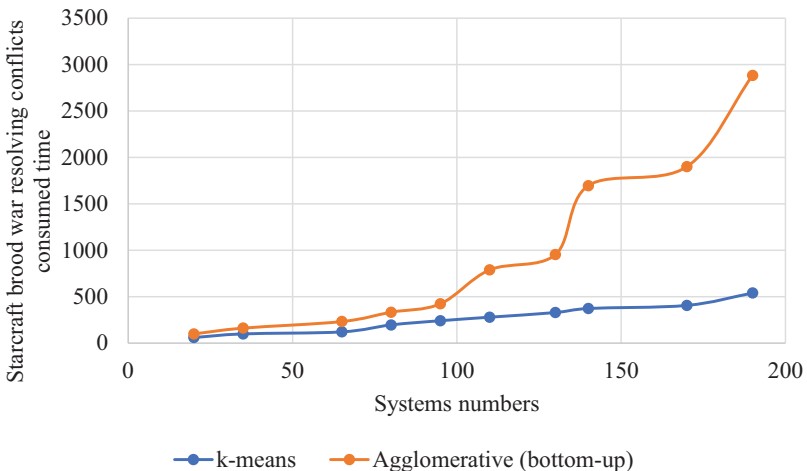

**Figure 17 StarCraft Brood War Time consumed in conflicts resolving during implementing SSBFCSoS using k-means vs. Agglomerative (bottom-up) clustering techniques.**

therefore, we need a clustering algorithm with the following features for the fusion into SSBFCSoS:

- Dealing with huge datasets such as k-means, CURE, BIRCH and CHAMELEON (*Sanse & Sharma, 2015*), STING (*Oyelade et al., 2016*), Agglomerative, and HASTREAM (*Nasraoui & N'Cir, 2019*). We display the data size criteria in the second column Table 11.

- Low time consumption. K-means is a very fast algorithm reducing the overall consumed time (*Nasraoui & N'Cir, 2019*).

- Not requiring special hardware as all chosen algorithms in Table 11.

- Scalability, the k-means algorithm is at the forefront of scalability as for SoS (*Sholla et al., 2017*).

- Accepting overlapping clusters, which is the effect on conflict solution. k-means is one of the algorithms that do not accept overlap between clusters (*Khanmohammadi, Adibeig & Shanehbandy, 2017*) as well as OPTICS (*Mirzaie et al., 2015*), HASTREAM and Liar Tree (*Nasraoui & N'Cir, 2019*), While algorithms C-means (*He et al., 2018*), SOM (*Sarlin & Eklund, 2011*) and K-Medoids (*Arora & Varshney, 2016*) accept overlapping. This comparison is in column four, Table 11.

- General implementation feasibility, as a trend in 2021 Artificial Intelligence utilises deep learning in clustering big data but that need proper hardware and cloud system, and applying to a certain application with the training dataset. K-means is easily applied to any general SoS application, and can easily be applied to devices with available medium capabilities as well. Finally, k-means is still one of the top five clustering algorithms (*Bangui, Ge & Buhnova, 2018*).

For these reasons, the k-means algorithm was the base of our proposed algorithm SSBFCSoS. We have compared k-means distance functions (Conclusions), then adopted a more appropriate specific measurement method as explained.

**Table 11 Comparison between the k-means algorithm and some clustering algorithms.**

| Algorithm | Dataset size | Scalability | Overlapping Clusters (Conflicts) | Time Consuming | Adapting with SoS/IoT |
|---|---|---|---|---|---|
| k-means | Huge | Yes | No | Low | Yes |
| K-medoids | Small | No | Yes | High | No |
| CLARA | Huge | Yes | No | Medium | Yes |
| Agglomerative (bottom-up) | Huge | Yes | No | High | Yes |
| STING | Huge | No | – | Medium | Yes |
| c-means | Medium | Yes | Yes | High | No |
| BIRCH | Huge | Yes | – | High | No |
| CURE | Huge | Yes | – | High | Yes |
| CHAMELEON | Huge | Yes | No | High | No |
| OPTICS | Small | No | No | Medium | Yes |
| HASTREAM | Huge | Yes | No | High | No |
| Liar Tree | Small | No | No | High | No |
| SOM | Small | No | Yes | High | Yes |

## CONCLUSIONS

Increasing the SoS capacity to accommodate more systems results in some issues among the SoS Components systems. Conflict is one of these issues. This paper focused on how to enhance the speed of detection and solution of conflicts that may arise while integrating new systems into an existing SoS.

We presented a method based on the use of the k-means clustering technique. Each cluster contains nearby systems according to pre-specified criteria. We can consider each cluster a Sub SoS (S-SoS) which in turn form the major SoS. We proposed the Smart Semantic Belief Function Clustered System of Systems (SSBFCSoS) which is an enhancement of the Ontology Belief Function System of Systems (OBFSoS). The proposed method proved the ability to detect and solve conflicts.

In order to test the applicability of the SSBFCSoS and compare its performance with other approaches, two well-known datasets were employed. They are (Glest & StarCraft Brood War). With each dataset, 15 test cases were examined. Considering that each testing was after every ten attempts of learning. To evaluate the proposed method, we applied it to homogeneous systems cases. The Glest game for a sample of about 70 systems divided into three clusters (S-SoSs). StarCraft Brood War was selected for a sample of about 100 systems divided into three clusters (S-SoSs). Furthermore, we have picked these cases specifically for the application because we intend to use homogeneous states.

Using clustering techniques showed better performance ratios. The proposed method (SSBFCSoS) demonstrated the time-reducing effect of SoS activities with the early detection of conflicts to be dealt with. We adopted an essential type of conflict, which is the struggle to obtain the optimal Decision for SoS. As a learned system, it avoids similar conflicts during attempts. It also helped to improve results and allowed us to deal with more CS in SoS with saving much time.

Two types of clustering algorithms have been tried; k-means and agglomerative (bottom-up). The algorithm k-means was adopted for SSBFCSoS for its better results, not only in terms of resolving conflicts but for the less time, it took as well.

In many cases, clustering results enabled the SSBFCSoS to check the conflict state while still handling conflict optimally. We achieved, on average, 89% in solving the conflict compared to 77% of the other well-known approaches. Moreover, it showed an acceleration of up to proportionality over previous approaches for about 16% in solving conflicts as well. It also reduces the frequency of the same conflicts by approximately 23% better than the other method, not only in the same cluster but even while combining different clusters. Eventually; the positive effect of the clustering process has appeared, the number of Component Systems (CS) has almost tripled compared to other methods while preserving integration and conflict resolution.

### Future work

The evaluation of this approach appears promising, but scalability issues remain to addressed. Consequently, future business trends should include developing more specific technologies to define conflict for larger quantities of systems and rules of conflicts. Besides, dealing with heterogeneous systems such as those found in smart cities. It can also be used as the classification method developed for different types of conflicts to provide orderly results to the user.

We used the k-means technique which depends on identifying the desired number of clusters before initiating the clustering process. We found that when some new systems are added to the already existing SoS, the system may belong to an S-SoS far from the one closest to it. That deviates us from the idea that we relied on in classification to some extent. This would prompt us to try different clustering techniques in the future.

Finally, the application on different numbers of systems within more than one method of clustering shows another issue; which is how to reach the ideal number of systems within each cluster.

Implementation showed that when the number of systems within one cluster reaches a specific number, the SoS becomes ideal in achieving its goals, which is evidently, the ideal number of clusters to achieve the SoS goal. This opens up another avenue for research on the same issue for future work. Besides, we will put a competitive study with Artificial Intelligence clustering techniques for the generalisation of our proposed method (SSBFCSoS) in the future as well.

### Funding
The authors received no funding for this work.

### Competing Interests
The authors declare that they have no competing interests.
## Author Contributions

- Eman K. Elsayed conceived and designed the experiments, performed the experiments, analysed the data, authored or reviewed drafts of the paper, and approved the final draft.
- Ahmed Sharaf Eldin Ahmed conceived and designed the experiments, authored or reviewed drafts of the paper, and approved the final draft.
- Hebatullah Rashed Younes conceived and designed the experiments, performed the experiments, analysed the data, performed the computation work, prepared figures and/or tables, authored or reviewed drafts of the paper, and approved the final draft.

## Data Availability

Codes are available in the Supplemental Files. The Supplemental Files show the code of the games systems that used in clustering techniques for SoS.

## Supplemental Information

Supplemental information for this article can be found online at http://dx.doi.org/10.7717/peerj-cs.468#supplemental-information.

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
