# Peer review of "Enhancing semantic belief function to handle decision conflicts in SoS using k-means clustering"

_PeerJ Computer Science, doi:10.7717/peerj-cs.468_

## Round 0.1 · original submission · Major Revisions

Please extensively revise the article in terms of experimental and analytical point of view, also please improve the language of the manuscript by a professional proof reader.

Reviewer 1 ·

Basic reporting

Use professional English in the text. For example, revision is required under the proposed system heading: The second phase presents the reversal of Glest's Java code generating into OWL Ontologies of all systems in each cluster. Plus merging the OWL ontologies for conflict detection and get a decision for each S-SoS.

Experimental design

Research question well defined, relevant & meaningful.

Rigorous investigation performed to a high technical & ethical standard.

Methods described with sufficient detail & information to replicate.

Validity of the findings

All underlying data have been provided; they are robust, statistically sound, & controlled.

Conclusions are well stated, linked to original research question & limited to supporting results.

Reviewer 2 ·

Basic reporting

No comment

Experimental design

Add more experiments

Validity of the findings

Variety of datasets should be used for validity of your proposed method

Additional comments

This manuscript propose the Smart Sematic Belief Function Clustered System of Systems (SSBFCSoS). SSBFCSoS consider an enhancement for method Ontology Belief Function System of Systems (OBFSoS). However, the paper, has more serious methodological flaws enumerated below.

*The implementation details should be clearly specified in terms of hardware and software used in the experiments.
* Each notation of all equations should be clearly described
* Compare your proposed method with other relative clustering algorithms. you just compare it with one algorithm .
* More datasets should be used for authentication of your proposed method.
However, despite many errors the text is simple and readable. Summarizing, although the overall concept is fine, the paper is very unprofessionally written and thus the reported results are very unconvincing. Therefore, I advise the Authors to add more experiments in your manuscript and resubmit it once again.

Reviewer 3 ·

Basic reporting

1. The English in the present manuscript is not of publication quality and require major improvements. Please carefully proof-read to eliminate grammatical errors.
2. Sufficient introduction and background are given.
3. Please add some more recent references.
4. The figures and tables must be explained in detail in the text. Explain what is going on in the figure and table. Figure 1 needs improvement. It is not easy to understand Figure 1.
5. Try to explain each step of the case study according to the proposed method. Apply and show the case study values in clustering techniques. Show the steps and results produced according to the algorithms used. e.g. where and how k-means is applied in the case study and what are the results produced.

Experimental design

1. If the quality of this article is improved, only then it will meet the aim and scope of the journal.
2. The knowledge gap investigated is not clearly justified. Add some more references to strengthen your problem.
3. The method needs improvement. The phases mentioned in the method do not give a clear picture and it is not easy for a reader to reproduce the result.
4. Mention the enhanced proposed form in the equation or algorithm. What is the final enhanced form? Defining Euclidean, Squared Euclidean, Manhattan, and Cosine distance measure is not enough. You need to explain at what stage you have applied what method.

Validity of the findings

Using clustering techniques require more sound statistical explanation. What results are produced with what clustering technique, statistically?

Additional comments

The article needs much improvement. The quality of the article is not very good but still can be improved. There are lots of grammatical mistakes. Try to work on it. It can be improved. Good luck.

---

## Round 0.2 · Minor Revisions

Dear authors

Thanks for the revised submission of your manuscript, the paper has been reviewed and found quite better than the previous version. However to polish it more please address the few comments of reviewer 2 and resubmit. Thank you

Reviewer 1 ·

Basic reporting

Clear and unambiguous, professional English used throughout.
Literature references, sufficient field background/context provided.
Professional article structure, figures, tables. Raw data shared.

Experimental design

Original primary research within Aims and Scope of the journal.

Validity of the findings

All underlying data have been provided; they are robust, statistically sound, & controlled.

Additional comments

All minor revisions are submitted.

Reviewer 2 ·

Basic reporting

The author has modified paper according to suggestions, and now the organization of the paper is improved and further required explanation has been provided.

Experimental design

The authors performed extensive experiments to test the performance of their algorithm . Experiment results are well elaborated. but I found some minor issues in the manuscript that need to be rectified like more clustering algorithms should be compare with the proposed method .

Validity of the findings

no comment

Additional comments

The author has modified paper according to suggestions, and now the organization of the paper is improved and further required explanation has been provided.

Although the language structure has been improved and authors were able to overcome grammatical mistake but it still needs to optimized.I suggest that authors should review the paper once more and optimize and repitative elements should be consolidated.

Reviewer 3 ·

Basic reporting

No comment

Experimental design

No comment

Validity of the findings

No comment

---

## Round 0.3 · Minor Revisions

A final check of the language shows that it is still not at a publishable standard.

Please carefully revise the language of the manuscript once more as it will be the last chance to polish your paper. thank you.

Please either have a colleague who is proficient in English and familiar with the subject matter review your manuscript, or contact a professional editing service to review your manuscript

Reviewer 2 ·

Basic reporting

Now this paper have incorporated the solution of all necessary requirements and they have worked on modifying the paper in the light of the recommendations. Format along with style of the paper is fine now.

Experimental design

The authors performed extensive experiments to test the their method . Experiment results are well elaborated.

Validity of the findings

The author has modified paper according to suggestions, and now the organization of the paper is improved and further required explanation has been provided.

Additional comments

it is a well written paper and suggested changes were made by authors. The article may be proof read from native English speaker to facilitate readers. it may accepted in current form.

---

## Round 0.4 · accepted · Accept

Thanks for revising the paper according to reviewer and editors comments.

Good luck